# "What do I think about implementing lung cancer screening? It all depends on how." Acceptability and feasibility of lung cancer screening in Australia: The view of key stakeholders about health system factors

**Rachael H. Dodd[1,2], Ashleigh R. Sharman[2], Henry M. Marshall[3], Mei Ling Yap[2,4,5,6], Emily Stone[7], Joel Rhee[8,9], Sue McCullough A. O. M.[10], Nicole M. Rankin[11]** *

1 The Daffodil Centre, A Joint Venture Between Cancer Council NSW and The University of Sydney, Faculty of Medicine and Health, The University of Sydney, Sydney, New South Wales, Australia, 2 Faculty of Medicine and Health, School of Public Health, The University of Sydney, Sydney, New South Wales, Australia, 3 University of Queensland Thoracic Research Centre and Department of Thoracic Medicine, The Prince Charles Hospital, Chermside, Queensland, Australia, 4 Collaboration for Cancer Outcomes, Research and Evaluation, Ingham Institute, UNSW Sydney, Liverpool, New South Wales, Australia, 5 Liverpool and Macarthur Cancer Therapy Centres, Western Sydney University, Campbelltown, New South Wales, Australia, 6 George Institute for Global Health, UNSW Sydney, Sydney, New South Wales, Australia, 7 Department of Thoracic Medicine and Lung Transplantation, St. Vincent's Hospital, University of NSW, Darlinghurst, New South Wales, Australia, 8 School of Population Health, Faculty of Medicine and Health, UNSW Sydney, Sydney, NSW, Australia, 9 Graduate School of Medicine, Faculty of Science, Medicine and Health, University of Wollongong, Wollongong, NSW, Australia, 10 TOGA Consumer Advisory Panel, Melbourne, Victoria, Australia, 11 Centre for Health Policy, Melbourne School of Population and Global Health, The University of Melbourne, Melbourne, Victoria, Australia

* nicole.rankin@unimelb.edu.au

## Abstract

### Background

Lung cancer is the number one cause of cancer death worldwide. Although international trials demonstrate that targeted screening using low dose computed tomography (LDCT) significantly reduces lung cancer mortality, implementation of screening in the high-risk population presents complex health system challenges that need to be thoroughly understood to support policy change.

### Aim

To elicit health care providers' and policymakers' views about the acceptability and feasibility of lung cancer screening (LCS) and barriers and enablers to implementation in the Australian setting.

### Methods

We conducted 24 focus groups and three interviews (22 focus groups and all interviews online) in 2021 with 84 health professionals, researchers, and current cancer screening program managers and policy makers across all Australian states and territories. Focus groups

**Data Availability Statement:** Data cannot be shared publicly because of privacy or ethical

restrictions. Public availability may compromise participants confidentiality or reveal confidential information about their employers. Reasonable requests for data may be sent to nicole. rankin@unimelb.edu.au or alternatively to The University of Sydney Human Research Ethics Committee (human.ethics@sydney.edu.au Project ID: 2020/743) or The University of Melbourne Human Research Ethics Committee (HumanEthics-Enquiries@unimelb.edu.au. Project ID: 25450.)

**Funding:** This study was financially supported by an Australian National Health and Medical Research Council (NHMRC) Ideas Grant (2019/GA65812) awarded to NMR. This study was also financially supported by a Metro North Hospital and Health Service (Queensland, Australia) Clinical Academic Fellowship and an NHMRC Investigator Grant awarded to HMM. The funders had no role in study design, data collection and analysis, decision to publish, or preparation of the manuscript.

**Competing interests:** The authors have declared that no competing interests exist.

included a structured presentation about lung cancer and screening and lasted approximately one hour each. A qualitative approach to analysis was used to map topics to the Consolidated Framework for Implementation Research.

## Results

Nearly all participants considered LCS to be acceptable and feasible but identified a wide range of implementation challenges. Topics (five specific to health systems and five cross-cutting with participant factors) identified were mapped to CFIR constructs, of which 'readiness for implementation', 'planning' and 'executing' were most salient. Health system factor topics included delivery of the LCS program, cost, workforce considerations, quality assurance and complexity of health systems. Participants strongly advocated for streamlined referral processes. Practical strategies to address equity and access, such as using mobile screening vans, were emphasised.

## Conclusions

Key stakeholders readily identified the complex challenges associated with the acceptability and feasibility of LCS in Australia. The barriers and facilitators across health system and cross-cutting topics were clearly elicited. These findings are highly relevant to the scoping of a national LCS program by the Australian Government and a subsequent recommendation for implementation.

## Introduction

Lung cancer is the leading cause of cancer death worldwide, [1] and has the highest cancer burden in Australia (18.6% of the total burden) [2]. In Australia, five-year survival is 17%, due to most patients presenting with late-stage incurable disease [2].

The findings of two large international randomised controlled trials, the National Lung Cancer Screening Trial (NLST) and the NELSON trial, have reported a 20–24% relative reduction in lung cancer mortality [3, 4]. Implementation of lung cancer screening (LCS) commenced in the United States (US) in 2013, with the US Preventive Screening Task Force recommendation updated in 2021 to lower the starting age of screening from 55 to 50 years and reduced smoking exposure from a 30 pack year history to 20 pack year history [5]. Other countries to have more recently implemented LCS for those at high-risk are Canada, with provincewide programs in Ontario [6] and British Columbia [7], while South Korea [8], Croatia [9], Poland [10] and Taiwan [11] have national LCS programs. LCS pilot programs have been launched worldwide, including in England [12], with ongoing trials in other jurisdictions (e.g., China [13], Brazil [14], New Zealand [15]). The outcomes of such real-world programs show promising outcomes in terms of detection of early-stage (e.g., 81.2% Stage I or II) disease and diagnostic accuracy (e.g., 2% false positive rate; invasive surgical investigations (0.6%) in those without lung cancer) [12] but significant variation in uptake in high-risk populations (ranging from 4% [16] to 52% [17]).

The Australian Government has invested in early scoping of a potential national LCS program [18], following an enquiry in October 2020 that concluded 'LDCT screening would enable unprecedented changes in clinical management and address poor outcomes for lung cancer that have been observed for years' [18]. This enquiry estimated that 580,000 Australians

would be eligible for screening on completion of a risk assessment tool, over 12,000 deaths would be prevented and of all screen detected lung cancer, over 70% would be diagnosed at an early stage. In October 2022, the Medical Services Advisory Committee (MSAC; an independent non-statutory committee of the Australian Government Department of Health and Aged Care that appraises and provides advice to Government on whether proposed medical services should be publicly funded) recommended funding a national program after considering the strength of the available evidence in relation to comparative safety, clinical effectiveness, cost-effectiveness and total cost, with Australian Government endorsement yet to be finalised [19]. The Australian healthcare system is a hybrid model. Citizens, permanent residents and refugees are given public insurance known as Medicare, but can also buy private insurance coverage and gain access to privately operated health facilities [20]. While the Australian Government determines whether new screening programs are implemented, state and territory governments are typically responsible for the planning and performance of screening program and healthcare facilities [21]. This results in complex policy decision making about how best to resource new screening programs.

International studies have identified complex barriers to LCS at participant, healthcare provider, organisational, health system and policy levels. These barriers include lack of awareness (of lung cancer and of screening), access challenges, cost concerns, fear (e.g. diagnosis, procedures) and stigma associated with smoking, lack of shared decision-making, scepticism from health professionals about evidence of benefits of LCS, as well as challenges in identifying people at high-risk [22, 23]. These have not been assessed in an Australian population. Pre-implementation data must be generated as: i) we cannot predict whether the same barriers will be relevant for the population, given the geographical spread and the complexities of the Australian health care system including disconnections between primary and tertiary care [24], ii) there may be additional barriers not previously documented for Australia's diverse population, both in terms of Indigenous (First Nations) communities and multicultural communities, and iii) we do not know how these factors may impact on the design and delivery of a LCS program. Further evidence is needed to understand whether LCS is viewed as broadly acceptable and feasible. Evidence is also needed about whether identifying the barriers and facilitators to LCS prior to implementation can facilitate the development of interventions and strategies to promote successful uptake in the Australian community.

Therefore, the aim of this study was to generate evidence about the acceptability and feasibility of implementing LCS in the Australian setting using an implementation science approach. The objectives were to gain an understanding of the perceptions and attitudes of health professionals about LCS implementation and to identify and analyse the potential barriers and facilitators using the original Consolidated Framework for Implementation Research (CFIR) [25]. These insights will help to develop the foundation for selecting LCS implementation strategies and help policymakers plan for potential LCS program implementation.

## Methods

### Participants

We invited participants to take part in focus groups, including general practitioners (GPs), primary care nurses, respiratory physicians, radiologists, oncologists, and other healthcare professionals from multiple disciplines, as well as current cancer screening program managers and policy makers. There were no specific exclusion criteria as we wanted to include professionals from multiple discipline and across Australia, and a passive snowballing approach was used, whereby focus group participants could recommend other colleagues to invite. No prior knowledge about LCS was required of participants.

### Recruitment and study processes

Participants were recruited across health professional groups including those practising in regional, rural and remote settings where lung cancer incidence is higher than in metropolitan settings. The study was advertised through various avenues such as newsletters of professional colleges, associations and organisations. We directly contacted state and territory cancer screening units and Primary Health Networks and when a response was received, we requested that study information be circulated via the organisation's communication networks. The research team members also shared study information on closed professional groups on Facebook. Access was made possible due to existing memberships of these groups via the authorship team.

Participants expressed interest in the study via an online invitation system (EventBrite) that contained dates and times for focus groups, as well as consent and information about the study. This information included who the researchers were, where they worked, and the aims of the research. Written consent involved participants downloading the consent form and emailing a signed copy to a member of the study team prior to the date of their focus group. A relationship was developed between the researchers and participants as a part of assigning them to a focus group. As participants nominated a focus group time that best suited their availability, a mix of professional disciplines were present within each group. In one instance, a focus group was specifically designated for one cancer screening program team. Participants completed a brief written questionnaire to collect demographic data prior to the focus group taking place. This study has been reported in line with the Consolidated Criteria for Reporting Qualitative Research (COREQ; see S1 File) [26].

### Focus group content

The focus groups were structured around a presentation developed by the research team, which included findings from international LCS randomised controlled trials, an overview of the Australian LCS enquiry, the proposed participant risk assessment tool (PLCOm2012), as well as international examples of LCS programs. The CFIR was used to develop a semi-structured moderator guide for the focus groups and explore the potential of LCS in Australia. This presentation was adapted between group events as it became evident what information was most important to cover and to reduce the time burden on participants.

### Data collection

All focus groups and interviews were conducted between February and July 2021. Each group or interview was moderated by a female research team member holding a PhD with expertise in behavioural science (RD, NR) and an interest in the feasibility of implementing a LCS in Australia. Twenty-two focus groups and three interviews were carried out via Zoom™ and two focus groups face-to face. Focus group duration ranged from 40 to 60 minutes, with a mean duration of 54 minutes. Three participants were not able to attend a focus group and so they each participated in individual interviews conducted via Zoom™. Participants were asked to freely express their thoughts regarding LCS and were offered a $100 gift card as reimbursement for their time.

### Data analysis

The focus groups and interviews were recorded via Zoom™, the audio file transcribed using TRINT™ and anonymised, and checked by one research team member for completeness. Three research team members (RD, NR, AS) independently familiarised themselves with three

randomly selected focus group transcripts and developed codes inductively which reflected the main topics from the discussions. From these transcripts, an initial coding framework was developed and a further six transcripts were discussed against this coding framework. All coding was discussed (RD, NR, AS), and any disagreements resolved before developing the final coding framework. All transcripts were then coded in NVivo (released in March 2020) by two research team members (RD, AS) using the final coding framework. The same team members subsequently mapped topics to the CFIR structure. This conceptual framework has been developed to 'guide systematic assessment of multilevel implementation contexts to identify factors that might influence intervention effectiveness' [27] and the selection of implementation strategies to overcome barriers and facilitators [28]. Constant comparisons were made between the researchers to assess consistency or differences and all disagreements were resolved by discussion with an additional team member (NR).

### Ethics approval

The study was approved by the University of Sydney Human Research Ethics Committee (2020/743).

## Results

A total of 84 participants took part in 24 focus groups and three individual interviews. A description of the sample is shown in Table 1. A graphic representation of the high-level topics in the coding framework is shown in Fig 1. The full coding framework is contained in S2 File. This manuscript focuses on health systems factors and cross-cutting topics. Separate manuscripts will report on stakeholder views about participation factors and smoking cessation. The coding framework is presented in relation to the CFIR domains and relevant constructs, as shown in Table 2 [29].

### Domain: Intervention characteristics

This domain relates to the intervention, that is, a LCS program and how it is perceived. The most relevant constructs within this domain were evidence strength and quality, relative advantage, adaptability, trialability, complexity and cost.

**Evidence strength and quality.** There was very strong support expressed by the healthcare participants for the implementation of LCS, particularly based on the evidence of improved outcomes from international trials.

*'I think there's a lot of interest in lung cancer screening based on the evidence shown by those two trials. In Australia, in Queensland they set up lung cancer screening . . . and they demonstrated that was reasonably feasible.' (FG1, Respiratory Physician) 'So, no I think it will be higher. It would be highly acceptable to patients and. And in terms of results, well, I think the results in terms of lives saved, lives extended, quality of adjusted life, years, etc., which is when the statisticians like to look at the outcomes of these things, will be far in advance of the other screening programs. (FG2, Thoracic Surgeon'*

We explored participants knowledge of LCS throughout the focus groups and while most had high level of knowledge, for some, the information presented was new.

**Relative advantage.** Participants recognised both the benefits and harms of LCS, but overall recognised the overarching evidence that showed LCS would detect early cancer which in turn leads to more effective treatment.

**Table 1. Sample characteristics.**

| Participant characteristics | Frequency (Percent) of Total n = 82* |
|---|---|
| **Age** | |
| 18–40 years | 36 (42.9) |
| 41–60 years | 37 (45.1) |
| 61+ years | 9 (11.0) |
| **Gender** | |
| Female | 48 (58.5) |
| Male | 34 (41.5) |
| **Country of Birth** | |
| Australia | 54 (65.9) |
| *Other* | *28 (34.1)* |
| **Aboriginal or Torres Strait Islander** | |
| Yes | 5 (6.1) |
| No | 77 (93.9) |
| **Country of University Education completion** | |
| Australia | 75 (91.5) |
| *Other* | 7 (8.5) |
| **Professional Role** | |
| General Practitioner (GP) | 13 (15.9) |
| Radiation Oncologist | 10 (12.2) |
| Nurse | 11 (13.4) |
| Radiologist | 9 (11.0) |
| Respiratory Physician | 9 (11.0) |
| Policy/Program Manager | 6 (7.3) |
| Medical Oncologist | 4 (4.9) |
| Allied Health Professional | 3 (3.7) |
| Researcher | 2 (2.4) |
| Trainee, GP Registrar | 1 (1.2) |
| *Other* | *14 (17.1)* |
| **Australian State or Territory of Work** | |
| New South Wales (NSW) | 36 (43.9) |
| Victoria (VIC) | 14 (17.1) |
| Queensland (QLD) | 11 (13.4) |
| Western Australia (WA) | 8 (9.8) |
| Tasmania (TAS) | 5 (6.1) |
| Australian Capital Territory (ACT) | 3 (3.7) |
| South Australia (SA) | 3 (3.7) |
| Northern Territory (NT) | 2 (2.4) |
| **Workplace Setting** | |
| Medical Centre/Community-Based Clinic | 11 (13.4) |
| Private Practice/Sole Practitioner | 8 (9.8) |
| Public Hospital | 33 (40.2) |
| Private Hospital | 2 (2.4) |
| Academic, University-Based Clinic | 4 (4.9) |
| Combination | 8 (9.8) |
| *Other* | *16 (19.5)* |
| **Practice Location** | |
| Urban/Inner-City | 43 (52.4) |

(*Continued*)

**Table 1.** (Continued)

| Participant characteristics | Frequency (Percent) of Total n = 82* |
|---|---|
| Suburban | 19 (23.2) |
| Rural | 8 (9.8) |
| Not Applicable (non-clinicians) | 12 (14.6) |
| **Nature of Practice** | |
| Private | 14 (17.1) |
| Public (Bulk-Billing) | 44 (53.7) |
| Non-Practising | 2 (2.4) |
| Not Applicable (non-clinicians) | 12 (14.6) |
| *Other* | *10 (12.2)* |
| **Years Worked Professionally** | |
| 0–10 years | 31 (37.8) |
| 11–20 years | 17 (20.7) |
| 21–30 years | 21 (25.6) |
| 30+ years | 12 (14.6) |
| Not Applicable | 1 (1.2) |

*Data missing for 2 participants

*'And the idea is that if you use a risk-based approach, then you can potentially improve the outcomes of the cancer. You pick up more patients with cancer and you'll have less false positives basically, and it'll be more efficient because you're selecting a higher risk group of patients to begin with.' (FG1, Respiratory Physician)*

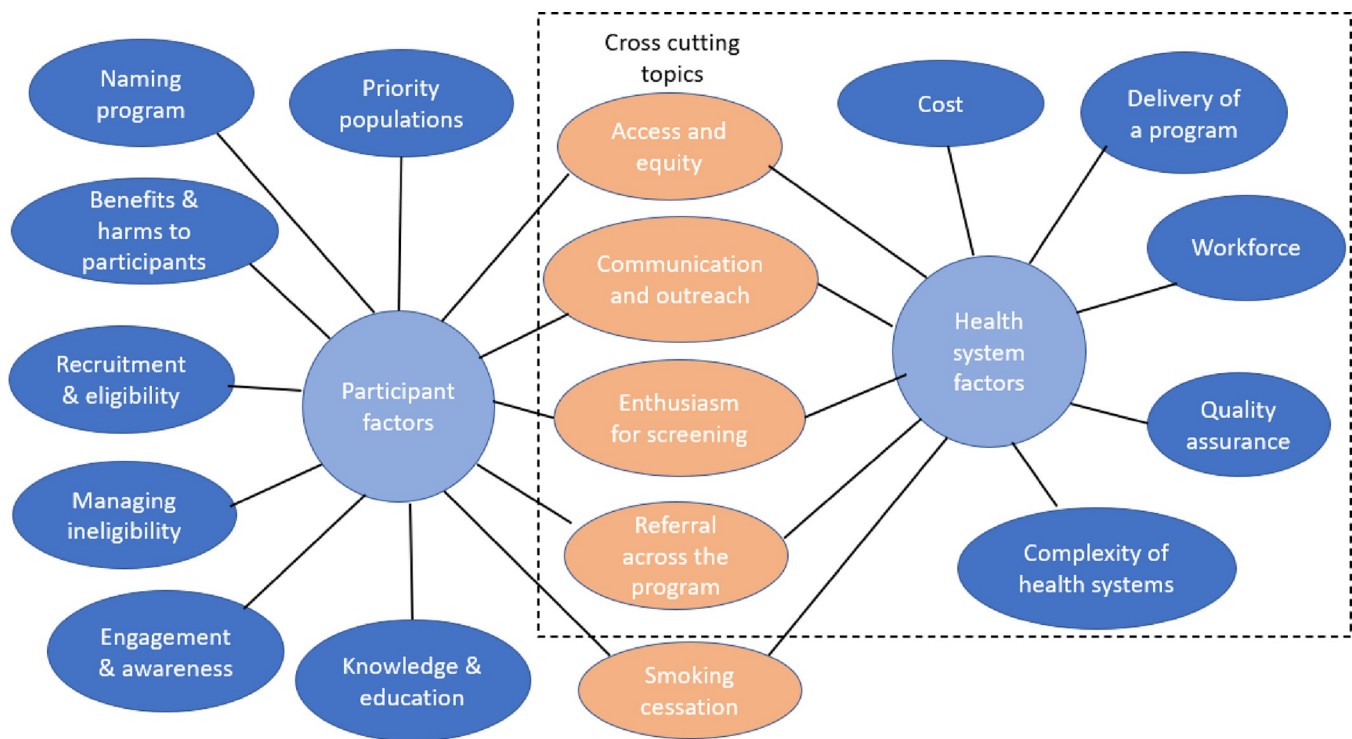

**Fig 1. Participant and health system factors identified (in blue ovals) during analysis including cross-cutting topics (in orange ovals), with the dotted line indicating those topics included within this manuscript.**

**Table 2. Mapping of participant and health system factor topics to CFIR constructs inclusive of CFIR descriptions.**

| CFIR Construct | | Short Description | Topics mapped to constructs |
|---|---|---|---|
| **I. INTERVENTION CHARACTERISTICS** | | | |
| A | Intervention Source | Perception of key stakeholders about whether the intervention is externally or internally developed. | |
| B | Evidence Strength & Quality | Stakeholders' perceptions of the quality and validity of evidence supporting the belief that the intervention will have desired outcomes. | Engagement & awareness (Participant factors) |
| C | Relative Advantage | Stakeholders' perception of the advantage of implementing the intervention versus an alternative solution. | Benefits and harms (Participant factor) |
| D | Adaptability | The degree to which an intervention can be adapted, tailored, refined, or reinvented to meet local needs. | Risk assessment (Participant factors) |
| E | Trialability | The ability to test the intervention on a small scale in the organization, and to be able to reverse course (undo implementation) if warranted. | Delivery of a program (Health System factors) |
| F | Complexity | Perceived difficulty of implementation, reflected by duration, scope, radicalness, disruptiveness, centrality, and intricacy and number of steps required to implement. | Complexity of health systems (Health System factors) |
| G | Design Quality & Packaging | Perceived excellence in how the intervention is bundled, presented, and assembled. | Naming the program (Participant factors) |
| H | Cost | Costs of the intervention and costs associated with implementing the intervention including investment, supply, and opportunity costs. | Cost (Health System factors) |
| **II. OUTER SETTING** | | | |
| A | Patient Needs & Resources | The extent to which patient needs, as well as barriers and facilitators to meet those needs, are accurately known and prioritized by the organization. | Access and equity (Cross cutting—Health Systems) Managing ineligibility (Participant factors) |
| B | Cosmopolitanism | The degree to which an organization is networked with other external organizations. | Screening and Assessment pathway (Health System factors) |
| C | Peer Pressure | Mimetic or competitive pressure to implement an intervention; typically, because most or other key peer or competing organizations have already implemented or are in a bid for a competitive edge. | Delivery of a program (subtopic: opportunistic screening) (Health System factors) |
| D | External Policy & Incentives | A broad construct that includes external strategies to spread interventions, including policy and regulations (governmental or other central entity), external mandates, recommendations and guidelines, pay-for-performance, collaboratives, and public or benchmark reporting. | Delivery of a program (Health system factors) |
| **III. INNER SETTING** | | | |
| A | Structural Characteristics | The social architecture, age, maturity, and size of an organization. | Delivery of a program (Health system factors) |
| B | Networks & Communications | The nature and quality of webs of social networks and the nature and quality of formal and informal communications within an organization. | N/A |
| C | Culture | Norms, values, and basic assumptions of a given organization. | N/A |
| D | Implementation Climate | The absorptive capacity for change, shared receptivity of involved individuals to an intervention, and the extent to which use of that intervention will be rewarded, supported, and expected within their organization. | Enthusiasm for screening (Cross cutting–HS factors) |
| 1 | Tension for Change | The degree to which stakeholders perceive the current situation as intolerable or needing change. | N/A |
| 2 | Compatibility | The degree of tangible fit between meaning and values attached to the intervention by involved individuals, how those align with individuals' own norms, values, and perceived risks and needs, and how the intervention fits with existing workflows and systems. | N/A |
| 3 | Relative Priority | Individuals' shared perception of the importance of the implementation within the organization. | N/A |
| 4 | Organizational Incentives & Rewards | Extrinsic incentives such as goal-sharing awards, performance reviews, promotions, and raises in salary, and less tangible incentives such as increased stature or respect. | N/A |
| 5 | Goals and Feedback | The degree to which goals are clearly communicated, acted upon, and fed back to staff, and alignment of that feedback with goals. | N/A |

*(Continued)*

**Table 2.** (Continued)

| | CFIR Construct | Short Description | Topics mapped to constructs |
|---|---|---|---|
| 6 | Learning Climate | A climate in which: a) leaders express their own fallibility and need for team members' assistance and input; b) team members feel that they are essential, valued, and knowledgeable partners in the change process; c) individuals feel psychologically safe to try new methods; and d) there is sufficient time and space for reflective thinking and evaluation. | N/A |
| E | Readiness for Implementation | Tangible and immediate indicators of organizational commitment to its decision to implement an intervention. | Workforce considerations |
| 1 | Leadership Engagement | Commitment, involvement, and accountability of leaders and managers with the implementation. | Workforce considerations–sub-topic: primary care education (Health system factors) |
| 2 | Available Resources | The level of resources dedicated for implementation and on-going operations, including money, training, education, physical space, and time. | Workforce considerations–sub-topic: primary care education (Health system factors) |
| 3 | Access to Knowledge & Information | Ease of access to digestible information and knowledge about the intervention and how to incorporate it into work tasks. | Workforce considerations–sub-topic: primary care education |
| **IV. CHARACTERISTICS OF INDIVIDUALS** | | | |
| A | Knowledge & Beliefs about the Intervention | Individuals' attitudes toward and value placed on the intervention as well as familiarity with facts, truths, and principles related to the intervention. | Priority populations (Participant factors) Enthusiasm for screening (Cross-cutting: see Health Systems) |
| B | Self-efficacy | Individual belief in their own capabilities to execute courses of action to achieve implementation goals. | |
| C | Individual Stage of Change | Characterization of the phase an individual is in, as he or she progresses toward skilled, enthusiastic, and sustained use of the intervention. | |
| D | Individual Identification with Organization | A broad construct related to how individuals perceive the organization, and their relationship and degree of commitment with that organization. | |
| E | Other Personal Attributes | A broad construct to include other personal traits such as tolerance of ambiguity, intellectual ability, motivation, values, competence, capacity, and learning style. | Knowledge and awareness (Participant factors) |
| **V. PROCESS** | | | |
| A | Planning | The degree to which a scheme or method of behaviour and tasks for implementing an intervention are developed in advance, and the quality of those schemes or methods. | Engagement and awareness Referral across the program (cross-cutting: Health System factors) |
| B | Engaging | Attracting and involving appropriate individuals in the implementation and use of the intervention through a combined strategy of social marketing, education, role modelling, training, and other similar activities. | Engagement and awareness (Participant factors) |
| 1 | Opinion Leaders | Individuals in an organization who have formal or informal influence on the attitudes and beliefs of their colleagues with respect to implementing the intervention. | |
| 2 | Formally Appointed Internal Implementation Leaders | Individuals from within the organization who have been formally appointed with responsibility for implementing an intervention as coordinator, project manager, team leader, or other similar role. | |
| 3 | Champions | "Individuals who dedicate themselves to supporting, marketing, and 'driving through' an [implementation]", overcoming indifference or resistance that the intervention may provoke in an organization. | Engagement and awareness (participant factors) |
| 4 | External Change Agents | Individuals who are affiliated with an outside entity who formally influence or facilitate intervention decisions in a desirable direction. | |
| C | Executing | Carrying out or accomplishing the implementation according to plan. | Delivery of a LCS program (Health System factors): subtopics—accreditation and physical infrastructure |
| D | Reflecting & Evaluating | Quantitative and qualitative feedback about the progress and quality of implementation accompanied with regular personal and team debriefing about progress and experience. | Quality assurance (health system factors) |

*'And in terms of results, well, I think the results in terms of lives saved, lives extended, quality of adjusted life, years, etc., . . . will be far in advance of the other screening programs.' (FG2, Thoracic Surgeon)*

**Adaptability.** When talking about adaptability, participants mainly spoke about how assessment processes to get people into screening could be adapted to meet local needs.

*'So . . . whether it is something sent out to them that's individualised, obviously advertising in that sort of thing and something mobile where they can see those sorts of places, they can go to get this done, really, I think is really helpful from my experience with different patients and different screening programs.' (FG10, General Practitioner)*

The most common suggestion was using mobile screening vans and bringing the screening to people. This was seen as very important to ensure access and equity, which was a recurring topic across nearly all groups.

**Trialability.** Participants suggested pilot testing on a small scale to begin with, starting first with those groups who would benefit most.

*'So, think of a way to sort of slow start it, if you like, so maybe start with the people who would benefit most. I mean, I think adding the patient population slowly, which also gives workforces a chance to expand to the increasing need in these kinds of things would be helpful.' (FG8, Pathologist)*

The main aim of pilot testing or gradual roll-out of implementation was perceived as learning 'as we go along' and establishing LCS to reach out to 'hard-to-reach' communities.

*'Starting up and getting things working on a small scale in order to get bigger. As we get more organised and aware and more intrinsically linked to the other areas, we need to go into, like the rural and the rural communities and organize the vans.' (FG6, Respiratory Nurse/ILST Coordinator)*

**Complexity.** Participants reflected on the complex health system in Australia, including public and private health systems, and federal and state government policies and funding arrangements.

*'And I think Australia, it's a problem because of the state and federal government divide . . . But for Australia, because of the federal state split, you know, it's not always that clear cut, which is a major problem, in my opinion.' (FG5, Medical Oncologist)*

The latter complexity was perceived as having been especially pronounced during COVID-19 and participants expressed concern that these complexities could have cost implications for a potential LCS program, as existing cancer screening programs have different management models (e.g., breast screening is a state/territory government operated, while cervical and bowel are federal).

Implementation barriers were noted to be complex and included program logistics such as ensuring a standard screening pathway, as well as deciding who will be responsible for each stage of the screening pathway. Other barriers included the complex health infrastructure and

access to CT scanners, delivery of the program and administration across health systems. A facilitator to implementation of LCS was nominated as engaging all stakeholders from the start to enable learnings to be shared across the states and for jurisdictions to not work in traditional 'silos'. Health professional education and planning activities were seen as key facilitators to overcome these silos.

*'Australia is very much siloed in this, 'oh we do the screening, but the report goes back to your GP'. . . And I think that really is a challenge for a screening program that you need to have buy-in and involvement from everyone. I think there's a lot of planning that needs to be done and a lot of collaboration [required].' (FG12, Respiratory Physician)*

**Cost.**    Across all focus groups, the biggest barrier to implementation of LCS was perceived to be the cost. Costs of LCS and particularly about the upfront costs of setting up a program, the outreach, and funding model were frequently discussed by participants.

*'And I guess the question is also who's going to pay for this? It's always down to the finances. Is the federal government going to pay for this? Is this going to be paid for by the state government? Where is that money coming from?' (FG8, Pathologist)*

Costs to set-up the program were seen to have been made less prohibitive as treatments (such as immuno- and targeted therapies) have become more expensive, but that there is the need for investment in the infrastructure and CT scanners. Regarding cost-effectiveness, participants spoke about the need to have a balance between the upfront costs and the long-term savings from LCS, perceiving the balance to be more in the favour of screening than not. Participants raised some questions about whether there was evidence behind the cost effectiveness of LCS.

*'I guess it's been unclear up until now whether there is a cost effective and efficient way to offer it to people.' (FG9, General Practitioner)*

**Domain: Outer setting.**    The outer setting domain examines influences external to LCS. Topics identified included access and equity, which were mapped to the CFIR constructs of patient needs and resources, cosmopolitism, peer pressure and external policy and incentives.

**Patient needs and resources.**    Needs of the patients were recognised through a strong emphasis on equitable access to LCS, particularly for those in regional and remote regions of Australia.

*'These are issues which will have to be thought about, particularly since primary lung cancer, I suspect, is going to become a disease of poorer people in regional areas in Australia as smoking rates continue to drop so that barrier to imaging might be quite significant.' (FG5, Radiation Oncologist)*

*'So, if participants have to pay to go to their GP and certainly in [our state], only a very small percentage of general practices actually bulk bill, there are significant out-of-pocket costs for people to see a GP. So, you get back less than half of what you have to pay [and] if that is the only means of entry into the program, for a participant to see a GP for a risk assessment to be performed and then get referred to the program, then the uptake and the acceptability, I predict will be very low.' (FG7, Director of Population Screening and Cancer Prevention)*

Access was frequently discussed and, that to facilitate implementation of LCS, a program has to be as easy and seamless as possible. Participants expressed a view that a centralised system of intake and referrals would improve access.

**Cosmopolitanism.** Participants discussed the need for LCS to link with local clinics and early assessment in hospital cancer services to ensure a smooth pathway across services. Working with private radiology clinics was perceived as important as they were thought to have greater capacity to conduct LDCT scans than public hospitals.

*'I don't think most people will be coming to hospitals for the LDCT that can be done anywhere at any radiology service.' (FG1, Respiratory Physician)*

**Peer pressure.** While a competitive advantage to implement LCS was not addressed in detail, some participants did mention that they already knew of some radiological practices offering opportunistic LCS. This included getting fee-covered diagnostic CT scans and private sector use of National Lung Screening Trial (NLST) criteria to select participants. The potential for competition between public and private radiology providers across a potential screening program was noted, as well as among other screening programs.

*'It is almost a competition between the various programs that people want to do prostate screening is lung nodule/lung cancer, breast, cervical screening program, and it's almost like you need a centralised screening, take out all the fundraising competition between them all.' (FG25, Radiologist)*

**External policy and incentives.** Participants highlighted the need for clear LCS guidelines and protocols, as well as accreditation to ensure quality for radiologists reading the scans. Some participants considered reimbursement for radiologists to become accredited to report on LDCT scans would be a significant incentive.

*'And perhaps I think if you left it open for any radiologists to do and had a Medicare rebate for it, you'd get every private practice in the country offering it. And that would just massively increase the places you could get it done.' (FG11, Radiologist)*

Using incentives as a potential method for recruiting potential LCS participants was also mentioned.

*'So perhaps when you were to roll it out the way you encourage participation, whether it is any incentives, do consider that, because how the research project officer has tried to increase recruitment was to give a. . . gift card to try things like that, to try to increase participation.' (FG1, Physiotherapist)*

**Domain: Inner setting.** The inner setting domain examines the organisational characteristics and context of potential LCS delivery, reflecting what the structural characteristics (are anticipated to be), implementation climate and readiness for implementation. As LCS implementation is not yet in place, some constructs had limited relevance in the analysis, as the organisational characteristics can only be anticipated rather than known.

**Structural characteristics.** Some of the barriers discussed by participants not familiar with the proposed LCS program, included not knowing how the program would be structured. The delivery of the program would require a great need to invest time in the infrastructure and data management of the program in order for the program to be successful.

*'We need to have a robust I.T. infrastructure to support it. We need a good database. We need to be able to do analytics, on how we're performing and do audits that we can fine-tune the program and make it more successful in the future.' (FG1, Respiratory Physician)*

*'And we've got some examples and I would suggest if you haven't already, you need to look at the three existing cancer screening programs because they're all structured very differently. We've got some centralised ones and more state-oriented ones, and they both have their pros and cons.' (FG11, Director of Cancer Screening)*

**Implementation climate.** This was mapped to the topic of enthusiasm for LCS. Overall, there was great enthusiasm from study participants, who perceived that their patients would also be enthusiastic, particularly when reflecting on the recent International Lung Screening Trial in Australia.

*'So I think from our experience here, it seemed like there was quite a lot of people who were willing to do this, and the uptake was quite good.' (FG4, Medical Oncologist)*

LCS was perceived as acceptable to both health professionals and their patients, with the key benefits being that lung cancer would be identified at an earlier, treatable stage.

*'As a clinician, I think it's acceptable and I'm assuming that [we] will get more early lung cancers for treatment.' (FG3, Radiation Oncologist)*

**Readiness for implementation.** This construct was particularly detailed, with most participants identifying a complex interplay of factors. The topics are grouped under the headings of 'workforce and program delivery' and 'primary care engagement and education'.

**Workforce and program delivery.** Participants expressed concern about the likely organisational barriers to implementation. This included the impact of a potential LCS program on the workforce, which professions would be most impacted, and a perceived need for extra resources. Many health professionals thought the system would become overwhelmed with the number of nodules detected on LDCT scans and incidental findings.

*'The other difference, I think, is radiologists who do breast screening they're all happy to stick a needle in and do the biopsy as well. But it's almost two different populations of radiologists that you need for lung cancer screening. You need the diagnostic monitoring, the interventional one less. It's a high-risk biopsy, a lung biopsy and. . . in my opinion, [we] probably shouldn't be doing them, except in major centres where you've got back up if something goes wrong, which is different to breast. I don't think we've discussed pathologists, but you need to increase multiple specialties.' (FG11, Radiologist)*

GPs and radiologist participants expressed a need to understand details of what workforce planning is required and whose role it is to 'do the screening'. The perceived risks of not preparing for a LCS program in advance included a system overload and long waiting lists for services.

Radiologists viewed LCS as significant to fit into an already busy workload and perceived a model similar to BreastScreen (the Australian Breast Cancer Screening Program), where a huge amount of radiology resources are dedicated to the program. One barrier to engaging the radiological workforce was that interpretation of chest LDCT scans is viewed as 'boring work' by some. To facilitate any resistance in the workforce, there was a perceived need for very cohesive relationships, with two different populations of radiologists needed: those who would interpret scans locally and a centralised team for review of clinically significant findings and

ensuring the diagnostic accuracy of LDCT scans. Other workforce challenges included limited access to respiratory physicians and surgeons in some geographical areas and a concern expressed about where additional workforce would be found.

*'Think[ing] about how to implement that in Australia and the resources required and plan for how you're going to deliver the program, how you're going to deal with the incidental findings or the nodules, who's going to follow them up, how is that going to be done and make sure you have capacity, particularly for the treatments, because as I say, there's only so many thoracic surgeons around that if you get your cancer diagnosed, but then you wait six months for your surgery, that's not very good.' (FG12, Respiratory Physician)*

Facilitators to LCS with regards to workforce planning included an increase in administrative staff across multiple specialties, timely referral for treatment, leveraging private practices around the bigger centres, set-up of major regional centres for scanning to avoid travel for patients, and the setup of eligible patient identification software before program rollout. Increasing the number of surgeons, public hospital theatre availability and oncologists in preparation for the diagnosis of more cancers was suggested, as well as clear referral pathways for dealing with incidental findings.

*Primary care engagement and education*: Participants viewed GPs as the person that controlled patients' access to LCS. Barriers discussed about GPs included concern for patients who did not engage with GPs (regularly or at all), not all GPs might be willing to discuss LCS, time challenges and a significant burden on GPs.

*'I'm talking about also this like in general practice of thinking about like the accessibility of referring people on . . . we've got all these false positives and that burden on the system, that's something I found that's commonly we have to think about in terms of trying to get access to surgeons or access to people.' (FG2, General Practitioner)*

Participants suggested that a practice nurse-led model would be a facilitator to alleviating some of these barriers.

Facilitators for primary care included: providing GP and nurse education about the evidence for LCS; pathways for recommendation to LCS and referrals; and brief online summaries/flowcharts of the screening and assessment pathway. Support in communicating the risks and benefits to participants and asking about smoking history to determine eligibility, was also seen as necessary. Another facilitator for primary care was perceived as integrating the risk assessment tool into medical software and setting up alerts on medical software to identify potentially eligible patients would be beneficial.

*'It's the practice software that can actually store lots of things and it can generate lists that practice nurses, practice nurse letters, inviting patients to come and chat about participating.' (FG22, Administrator)*

### Domain: Characteristics of individuals

**Knowledge and beliefs (about the intervention).**   We have focused our interpretation on characteristics of individual clinicians, rather those of the LC screening participants. Most focus group participants were familiar with the evidence behind LCS and had positive attitudes towards LCS.

*'Yeah, I mean, look, I mean, I'm obviously fairly biased in that regard, but I mean, there's little doubt that the evidence will bear out that it's far more cost effective than breast cancer screening in my view.' (FG2, Thoracic Surgeon)*

*'If the early evidence shows that there's a benefit, then I'm all for it as a radiologist.' (FG20, Radiologist)*

Most reported being confident in their ability to carry out LCS.

**Other personal attributes.** Motivation and capacity of health professionals were both recognised as two important attributes that would facilitate the success of LCS, including the role of radiologists in interpreting scans.

*'It can be a mind numbingly boring challenging to remain motivated to try to screen 100 CT chests and I don't think we've worked out yet how to do that' (FG25, Radiologist)*

## Domain: Process

**Planning.** Across the focus groups, participants expressed the importance of thoroughly planning the program and for processes to be in place before LCS was (potentially) implemented. This included the need for a detailed screening and assessment pathway to ensure those screened are referred to the appropriate services.

*'Think about how to implement that in Australia and the resources required and plan for how you're going to deliver the program, how you're going to deal with the incidental findings or the nodules, who's going to follow them up, how is that going to be done and make sure you have capacity.' (FG12, Respiratory Physician)*

*'Like if we have the structures in place and everything, then it will be easier type of thing to do.' (FG2, General Practitioner)*

In almost every focus group, it was noted that program delivery would benefit from harnessing lessons from other screening programs, including cancer and cardiac rehabilitation programs. Participants stated the need for very clear messaging from program commencement, not staggered age criteria rollout like the National Bowel Cancer Screening Program, with the need to build on past lessons. LCS was most likened to BreastScreen with the use of mobile 'pop up clinics' to reach those people who do not attend primary care. Participants thought LCS could follow the BreastScreen model of self-referral, data and imaging, and quality control protocols. A need to learn what *has not* worked in some of the cancer screening programs and to draw on expertise of those involved in previous implementation, including how to de-implement ineffective practices were also identified. Some participants thought lessons were also to be learned from the COVID pandemic, including the use of telehealth.

**Engaging participants.** This construct covered cross-cutting topics of communication and outreach. Patient and professional champions who are passionate leaders were perceived as facilitators to reaching out to both patients and (healthcare) colleagues to build enthusiasm for LCS.

*'So . . . GP champions or something like that because, you know, half our patients in this area come from a non-English speaking background.' (FG3, Radiation Oncologist)*

*'It's got to be recruiting your clinician champions and what I mean clinicians, I don't just mean general practitioners. I mean whatever entry level screening health so, professionals that are going to be promoting the screening.' (FG6, General Practitioner)*

Participants reflected on the need to have communication strategies about a LCS program ready to share with potential participants. Strategies were also thought to be needed about how

best to reach those populations most at risk, through education and marketing to reach both the whole population, as well as the priority populations.

**Executing.** This construct mapped closest to 'delivery of the program' and the cross-cutting 'access and equity' topics. Delivery was considered to need agreed protocols and a screening and assessment pathway, as well as consideration of logistics and infrastructure. To carry out LCS according to plan, some participants thought the Population Based Screening Framework [30] should be utilised including structures for a follow-up pathway. One facilitator for implementing a LCS program was to use a risk assessment tool, with participants suggesting review of an existing 'heart calculator' tool. In terms of managing incidental findings, participants highlighted experiences from bowel screening, where colonoscopies can find other bowel conditions that require surveillance or treatment. Participants highlighted the need for the screening and assessment pathway to be based on a participative approach to encourage uptake. Telehealth, a consultation with a healthcare provider by phone or video call, was offered as a strategy for managing patients across the screening pathway.

> 'And I think certainly with COVID-19, a lot of hospitals have moved to more telephone consultations. And I think this kind of model could be very good for lung cancer screening. So, we might have to look at a paradigm shift in how we manage these patients.' (FG1, Respiratory Physician)

Robust IT infrastructure and data management were perceived as other vital components of delivery. A key barrier to implementation was the perception that rural and remote areas of Australia would not have enough CT scanners for a program to be feasible. Facilitators included: the use of a patient portal, electronic or web-based delivery of services (e.g., initial referrals and appointment bookings), the ability to conduct practice audits, robust reporting and documentation of LDCT scan results, with reporting to use computer-aided design to increase sensitivity or increase speed of interpretation. Participants particularly wanted access to previous CT images wherever a LCS participant got a scan, reducing the need to return to the same clinic, and thereby increasing patient choice. Facilitators included systems to monitor and support people across the program and to enhance compliance for follow-up scan; these were considered as markers of a successful program.

Other 'delivery of the program' facilitators included logistics, such as utilisation of existing services like private radiology providers, to not commencing the program until having proven capacity to do so, and to set participant's expectations through developing quality patient education and information.

Focus group participants strongly supported executing a model like the United Kingdom (UK) 'Lung Health Check'. This model was perceived by participants to feature: a clear screening and assessment pathway to target people; having very good strategies in place to implement at community level, enabling self-referral, risk assessment conducted by nurses; people being able to access mobile screening vans to avoid stigma associated with attending a hospital when they are not unwell; and the convenience of mobile screening vans.

> '*With some communities, it might actually be being able to provide it in a supportive environment where people were able to attend and take part as part of their participation in a community morning tea program and access to tea and biscuits and so forth.*' (FG26, Chief Executive)

> '*We [are] in. . . a town of five thousand people, we serve about 10,000 people locally. We have the Breast Screen coming once a year. We have osteoporosis access van coming once in two years. And I don't think there's any reason why a lung cancer bus can't come in and park in front of our library and have a low-dose CT done. Yeah, that's a really good idea.*' (FG9, General Practitioner)

**Reflecting and evaluating.** The 'quality assurance' (QA) topic mapped most closely to this construct. A robust QA program was considered a key factor to program success. QA was loosely defined, with most participants referring to monitoring of software and imaging systems and also to the interpretation of scans and the use of artificial intelligence (AI). AI was seen to facilitate finding nodules and look for incidental small lung lesions in the future as an aide rather than solely. The perceived advantages of AI were its ability to automate tedious tasks, and that scans performed in rural and remote centres could be centrally assessed as a means of cost reduction. Further content for this topic is grouped under two headings: accreditation and physical infrastructure.

*Accreditation*: Having accreditation for reading images was discussed by many participants. This was perceived as part of QA for radiologists, with a dedicated subgroup skilled in reading and reporting of CT images.

'So whether they want to do that, you have to have certain criteria for being a reporter of lung screening, like BreastScreen has that, you have to be breast screening accredited.' (FG5, Radiologist)

Barriers included a (potentially) reduced number of radiologists able to perform reporting as there would need to be radiologists dedicated to a LCS program. Many participants thought this decrease would be quick to overcome once implementation started, increasing quality and motivation for radiologists to be involved in LCS.

**Physical infrastructure.** Participants thought that CT scanners and CT images needed to be high quality, with strict guidelines to maintain consistency in reporting. Also important was how best to manage transfer and storage of images and what standard recommendations will be place. Participants viewed this aspect of QA as essential for program success. The multidisciplinary team was nominated as a facilitator of a QA measure.

## Discussion

This qualitative study explored the acceptability and feasibility of a potential LCS program in Australia from the perspective of healthcare providers, with a particular focus on the implementation barriers and facilitators at the health system level. This is the first Australian study and one of the most comprehensive international studies to consider health system factors prior to the commencement of LCS implementation. Using the CFIR framework facilitated analysis of qualitative data and topics synthesis to identify the barriers and facilitators most relevant to LCS implementation in Australia.

We identified key constructs of readiness for implementation, planning and executing a LCS program. The 'process' domain and the constructs of planning, engaging, executing and reflecting and evaluating must be viewed through the lens of 'pre-implementation' planning and evidence gathering. The focus group topics showed the numerous factors that need to be addressed to enable successful program planning and execution, whilst also planning for the factors that will enable future program evaluation, such as quality assurance. This contrasts with findings from a recent systematic review of US-based LCS programs [31] to describe the barriers and facilitators to LCS implementation in the US setting. The review identified that the CFIR constructs of external policy and incentives (outer setting domain) and executing (process domain) were the most common health system factors, while at the provider level, evidence strength and quality (intervention characteristics domain) were most salient. The potential persuasive nature of financial incentives directed at participants or providers need to be considered alongside their impact on participants making a fully informed shared decision

with their health professional about taking part in LCS. Our study shows how implementation constructs change in relevance according to context and stage of adoption, with far greater emphasis placed on readiness and planning of processes in the pre-implementation phase in our findings.

Study participants showed strong support for implementation of LCS and thought that the workforce would be willing to engage in a program underpinned by quality evidence available from large-scale international trials. Provider scepticism regarding the evidence for LCS has previously been identified as a barrier to LCS in the US [22]. However, subsequent to the NLST, robust evidence from trials including NELSON [3, 4], pilot studies and real-world program data have demonstrated the mortality benefits and identification of lung cancer at an early stage [12, 32]. This evidence base should help Australian healthcare providers to see the relative advantage of introducing a LCS program.

Participants in our study identified that a great deal of investment and planning (in the 'process' domain) was needed prior to implementation if a program is to succeed, which has been previously observed in US settings [29]. Gesthalter and colleagues [33] found that of three Veteran Administration sites in which LCS was implemented, the site with the most carefully planned and facilitated program was the most successful at incorporating recommended elements of LCS. This site used a team model of sharing best practices and learning from each other's experiences. Our study corroborates these findings, with participants emphasising a need to share lessons and avoid working in silos. Centralised systems for intake, referral and review of radiological findings were promoted by study participants as elements of likely success, as seen in US programs [34]. The potential use of AI was discussed to aid the reading of scans while easing the workload pressure and being able to automate tedious tasks. Research in the area of AI and cancer screening is rapidly evolving, with a study demonstrating improved sensitivity and specificity when using of a 'decision-referral' approach to reading mammograms, combining the strengths of AI and radiologists [35]. AI has also been shown to perform favourably when predicting risk of malignancy in pulmonary nodules [36]. If AI is to be used in a potential LCS program, future research will be required to evaluate which approach will be most effective and meet quality assurance standards.

The cost of LCS remains an important concern for many countries contemplating program implementation and participants in many of the focus groups reflected on this topic. The topic of cost links with that of access and equity for the Australian setting. The distribution of the Australian population across wide geographical areas and a disproportionate burden of lung cancer on First Nations [37] communities meant that discussions focused on reducing costs for both potential participants and providers, in order to maximise recruitment. Strategies suggested as a solution to access challenges have included travel vouchers, shuttle services, and offering appointments out of hours [22]. Study participants perceived the cost-effectiveness ratio to be likely favourable and that cost-effectiveness would be achieved if those at high risk take part in a LCS program. The need to monitor participation from high-risk groups must be monitored from the start [38].

Participants flagged the complexity of LCS ('intervention characteristics' domain) and that of the health system as potentially impacting on multiple stages of the screening and assessment pathway. Discussions highlighted that in the initial program establishment, it should be made explicit how delivery of a screening program across both public and private health systems would be managed, as well as the responsibility across federal and state governments having to run a potential program. Participants placed very strong emphasis on the learnings from existing screening programs in cancer and other chronic conditions. For example, the national breast cancer screening program BreastScreen does not rely upon people to access a GP to arrange the screening test or arranging further investigations.

There was support for gradual implementation of LCS in the study findings, similar to the approach used in the UK [38], which would allow a pilot program to be adapted where needed, with tailoring of strategies to address potential barriers and potential solutions across multiple stages of implementation. Pilot programs and trials have been run in the UK (e.g., Lung Screen Uptake Trial [39], early detection of lung disease pilot [40] UKLS), as well as the International Lung Screening Trial across Australia, Canada, Hong Kong, UK and Spain [41]. These trials and pilot programs have enabled various components of the screening and assessment pathway to be tested, such as invitation methods, targeted approaches to identifying those at high risk, use of the validated $PLCO_{m2012}$ risk assessment tool and nodule management protocols. Such outcomes will help determine how best to engage and retain people at high-risk and what further work is required to customise recruitment and risk assessment, as well as management of incidental findings [41].

Enthusiasm for LCS ('inner setting' domain) was consistently expressed by study participants and they anticipated similar enthusiasm would be likely for the general population. However, this enthusiasm was tempered by the emphasis placed on investment in planning prior to implementation. The main facilitators highlighted by participants at health system and policy levels were developing clear guidelines and policies for all stages of the screening and assessment pathway, a centralised system to manage scan results and utilising existing screening services. At the participant and provider level, education and training particularly for general practice and radiologists, and need for patient and professional champions were identified, which is consistent with previous studies [34, 42]. Local champions have been key facilitators to LCS implementation in the US [34, 42], with findings that a 'bottom up' approach, gaining buy in and input from participants and frontline staff responsible for implementation, was more effective than a 'top down' approach [29].

The study findings highlight concerns that challenges for the workforce could result in poor LCS program outcomes if not addressed. This included long waiting periods for participants and loss to follow up in the program, should there be a loss of trust from patients when the health system does not operate smoothly. Such potential risks need to be managed alongside the design and development of robust IT infrastructure and data management. Lessons can be learnt from other cancer screening programs in Australia, as well as other LCS around the world, with the US the furthest in the implementation process. Execution and evaluation of LCS ('process' domain) are key systems need to be put in place to enable ongoing evaluation of QA and clinical outcomes.

## Strengths and limitations

This qualitative study included a diverse sample of health professionals across all eight states and territories in Australia who would be responsible for future implementation of LCS. We engaged participants from a wide range of professional disciplines and geographical locations to capture a wide range of perspectives. However, we may have attracted more participants who were knowledgeable about LCS than not due to the nature of recruitment for studies around a particular topic, and we did not formally measure their knowledge prior to focus groups taking place. Therefore, those participants with more knowledge may have felt more comfortable speaking to the feasibility and acceptability than those who did not. However, valuable and new perspectives were gained from those who had no knowledge of LCS prior to the study. A further strength was the use of a formal implementation framework to shape the interview guide and data analysis of topics. We acknowledge that the resulting data do not reflect the perceptions of LCS participants. Our team has recently published a study of Australian participants in the International Lung Screening Trial, which found that individual

motivation to screen must be accompanied by strategies that enable opportunities and enhance individual's capabilities to screening for lung cancer [43]. Furthermore, our team is currently undertaking research to explore patient perceptions of barriers and facilitators to LCS program in focus groups with a diverse range of culturally diverse community members.

## Conclusions

This study has comprehensively identified that LCS implementation is acceptable and feasible to healthcare professionals in Australia. The health system factors of relevance to implementation were thoroughly explored using the CFIR implementation framework. We identified a wide range of barriers and facilitators to implementation, which can help to guide the selection of strategies to facilitate implementation. A need for careful planning, consideration of access and equity issues alongside a motivated and educated workforce will help to enable a potential LCS program in Australia and ultimately improve health outcomes for people diagnosed with lung cancer.

## Supporting information

**S1 File. Completed COREQ Criteria Checklist for manuscripts.**
(DOCX)

**S2 File. Coding framework: Generation of topics across all factors.**
(DOCX)

## Acknowledgments

We thank the focus groups participants and those organisations who willingly enabled us to contact their membership.

## Author Contributions

**Conceptualization:** Rachael H. Dodd, Henry M. Marshall, Mei Ling Yap, Emily Stone, Joel Rhee, Sue McCullough A. O. M., Nicole M. Rankin.

**Data curation:** Rachael H. Dodd, Ashleigh R. Sharman, Nicole M. Rankin.

**Formal analysis:** Rachael H. Dodd, Ashleigh R. Sharman, Nicole M. Rankin.

**Funding acquisition:** Emily Stone, Joel Rhee, Nicole M. Rankin.

**Investigation:** Rachael H. Dodd, Nicole M. Rankin.

**Methodology:** Rachael H. Dodd, Nicole M. Rankin.

**Project administration:** Rachael H. Dodd, Nicole M. Rankin.

**Resources:** Nicole M. Rankin.

**Supervision:** Nicole M. Rankin.

**Visualization:** Nicole M. Rankin.

**Writing – original draft:** Rachael H. Dodd, Henry M. Marshall, Mei Ling Yap, Nicole M. Rankin.

**Writing – review & editing:** Rachael H. Dodd, Ashleigh R. Sharman, Emily Stone, Joel Rhee, Sue McCullough A. O. M., Nicole M. Rankin.

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
