## [Decision Letter · Decision Letter 0]

21 Dec 2022

PONE-D-22-30637"There's a tension between what is most feasible versus understanding equity of access.” Acceptability and feasibility of lung cancer screening in Australia: the view of key stakeholders about health system factorsPLOS ONE

Dear Dr. Rankin,

Thank you for submitting your manuscript to PLOS ONE. After careful consideration, we feel that it has merit but does not fully meet PLOS ONE’s publication criteria as it currently stands. Therefore, we invite you to submit a revised version of the manuscript that addresses the points raised during the review process.

We look forward to receiving your revised manuscript.

Kind regards,

Jason Scott

Academic Editor

PLOS ONE

Journal Requirements:

Additional Editor Comments:

No further comments to add; the reviewers haven't identified any major issues, instead they are largely requests for further clarification and contextualisation to help readers, particularly an international audience. 

Reviewers' comments:

Reviewer's Responses to Questions

**Comments to the Author**

1. Is the manuscript technically sound, and do the data support the conclusions?

Reviewer #1: Yes

Reviewer #2: Yes

2. Has the statistical analysis been performed appropriately and rigorously? 

Reviewer #1: N/A

Reviewer #2: N/A

3. Have the authors made all data underlying the findings in their manuscript fully available?

Reviewer #1: No

Reviewer #2: Yes

4. Is the manuscript presented in an intelligible fashion and written in standard English?

Reviewer #1: Yes

Reviewer #2: Yes

5. Review Comments to the Author

Reviewer #1: Overall: Thank you to all authors for this interesting paper. It is of great importance to the cancer screening research. The framework provides a clear method of analysis and the findings are useful for the readers to understand how acceptable HCPs think LCS can be implemented and methods to aid the implementation.

Abstract:

Pg 2 – “Focus groups were mostly held via videoconferencing” – how many were held over this method? I am guessing the others were in person could you amend and clarify?

Pg 2 – were all participants from the same region or across Australia?

Pg 2 – “Participants took part in facilitated discussions of about one hour per group” – perhaps reword and state “Focus groups lasted approximately one hour each” to be more clear.

Introduction:A well written introduction, that clearly documents LC prevalence, the important of screening and the need for this research.

Pg 3 – “LCS pilot programs have been launched worldwide, including in England,”, you may discuss this later on but maybe indicate the results of the research undertaken in England on LCS

Pg 3 – missing bracket - “jurisdictions (e.g., China,12 Brazil,13 New Zealand14.”

Pg 4 – the justification for using the Consolidated Framework for Implementation Research (CFIR) might this be best to have in the methods rather than in the introduction?

Methods:

Throughout section: Throughout the methods could you indicate which authors completed which activity using for example “Each group was moderated by a researcher with expertise in behavioural science (Initials of person).” This can be really helpful for readers to know who lead the focus groups, who moderated or did anyone make notes during the focus group, who was involved in the analysis, who conducted the interviews.

Throughout section: It is good practice to report qualitative studies in line with the Consolidated criteria for reporting qualitative research (COREQ) by Tong et al 2007 (Consolidated criteria for reporting qualitative research (COREQ): a 32-item checklist for interviews and focus groups). Could you please use and add the necessary sentence into the methods.

Participants: No comments.

Recruitment and study processes:

Pg 5 – Consent – was this written or verbal? Was this collated prior to the date of the focus group?

Pg 5 – “The research team members also shared study information on closed professional groups on Facebook” – how did you gain access if they were closed? Did you ask a gatekeeper or someone who was a member to circulate information?

Focus group content:

Pg 5 – could you consider providing the semi-structured guide as a supplementary file? Or providing example of the questions/prompts within the guide.

Data collection:

Pg 5 – “All focus groups were conducted between February and July 2021 and groups were mixed by professions.” – you have already highlighted in the recruitment that the focus groups had a mix of people (except one) maybe remove the end of this sentence.

Pg 5 – “Groups were mostly carried out via Zoom” – guessing others were face-to-to face. Could you indicate how many were over zoom and face to face.

Pg 5 – “lasted no longer than one hour” could you report the range of length and the mean.

Pg 5 – “three individual interviews were conducted” – you might consider adding this into the abstract as focus groups were not the only method of data collection completed.

Pg 5 – “three individual interviews” – were the three interviews conducted over zoom too? – as with comment above could you indicate who completed.

Pg 5 – “$A100”- remove the A

Pg 5 – Did the same people lead the focus groups? I am assuming so by “moderated by a researcher” – as with comment above could you indicate who completed. Did the moderator or another member of the research team take notes/observations during the focus groups? If so, were they used in the data analysis.

Data analysis:

Pg 5 – “The focus groups were recorded” – was this just audio, or video too? Were the three interviews recorded?

Pg 5 – “familiarised themselves with three transcripts” – were the focus group transcripts or interview or a mixture. Can you clarify in the paper

General comment – the steps of data coding is very clear. Thank you for this!

Results: General comments: I am not familiar with the CFIR. Therefore I am sorry if any of the below comments are not applicable.

Table 1: the subcategories under the main heading, could you perhaps indent the categories slightly to make it easier for the reader to differentiate between the heading and sub categories.

Table 1: There seems to be data missing. For example, under ‘Aboriginal or Torres Strait Islander’ can you report the no and report the other countries of university education of the remaining 5 (given there are 2 sets of missing data). Perhaps have an other category as you have for ‘Country of birth’?

Table 2: Having looked into the CFIR constructs, seems to be missing some characteristics, I can see from the references there is a 2009 reference but when I have looked there is an updated CFIR constructs. Could you possible make it clear if you are using the original framework or updated (additional file 6 - https://implementationscience.biomedcentral.com/articles/10.1186/s13012-022-01245-0)

Table 3: Thank you for providing a quote table. This is really helpful. I particularly like that you have provided more than one quote from different focus groups and participants. Could you perhaps do this for the domains where only one is present?

Domains: While I like the table with the quotes in, I think integrating the quotes into the narrative itself would be beneficial to support your interpretations and can also help the reader understand some of the narrative for example “Participants raised some questions about whether there was evidence behind the cost effectiveness of LCS.” – what were the questions raised? I appreciate that word count will be difficult and the amount of the domains and sub-categories that are being reported.

General comment: Your results are really interesting.

Discussion:

Pg 23 “one of the most comprehensive internationally studies” – international?

Pg 25 “Breast Screen does not” – breast cancer screening?

Pg 27 – thank you for acknowledging the limitation that this paper does not involve LCS participants, but signposting the reader to your paper. This is really helpful and provides a balance on the implementation pf LCS in Australia.

Figure 1: Can you highlight what the orange bubbles mean in the figure? You have highlighted what the dotted line is but the a key of the colours would be useful.

Reviewer #2: Overview

A topical paper with some interesting findings, which will be key in understanding whether there is a place for lung cancer screening in Australia, as well as more generally worldwide. Of particular interest are repeating participant concerns relating to test specificity and patient pathways, e.g. recall rates, incidental findings, funding, access, and the impact of referral and treatment on current systems.

I am not familiar with the healthcare provision in Australia and would have liked some brief scene setting as to the structure of the different funding and commissioning streams.

I like the idea of a quote in the title of the paper, but I’m not sure I understand the one that’s been chosen.

I think it is important to include in your introduction some data from diagnostic accuracy studies of Computed Tomography (CT) and Low Dose Computed Tomography (LDCT), as well as a basic explanation of the two different tests. Include information about radiation risk, and why it might be important for a test to be ‘low dose’ in a screening population, discuss risk versus benefit and why high-risk populations only are currently being screened. Also, include any health economics and cost-effectiveness data published (if there is any) about the cost effectiveness of Lung Cancer Screening (LCS) programmes.

Introduction

A nice overview is provided of the worldwide status of LCS programmes in various countries, but I would be interested to see more detail around this, such as are they using CT or LDCT, how often, in which populations, is there published data regarding outcomes etc.

The third paragraph mentions a recommendation by ‘the Medical Services Advisory Committee’, some context might help explain who they are and why they made this recommendation.

Some interesting findings from other international studies are mentioned in the next paragraph, with barriers such as ‘fear and stigma’, can you explain these a little more. Fear and stigma of what? ‘Scepticism about evidence of benefits’, who is sceptical? Are these worries justified? Diagnostic accuracy figures could be used to illustrate how effective an LCS programme can be in the correct population. At the end of this paragraph the sentence beginning, ‘Further evidence is needed to understand whether LCS …’ is a long and complex sentence that is difficult to read and understand, the authors could consider splitting it up to make it clearer.

Paragraph four, which outlines the aims and objectives and introduces the ‘Consolidated Framework for Implementation Research’, feels a little confused. Consider moving the explanation of the framework to the methods data analysis section, sentence beginning, ‘This conceptual framework has been developed to…’.

Methods

The results section felt a little repetitive at times, perhaps due to the mapping of results into the CFIR framework, justification for the use of this framework in this section would help.

In ‘participants’, the authors state ‘there were no specific exclusion criteria’, explain why not and how participants were selected instead. Edit: a ‘passive snowballing approach’ is mentioned later, it might be better to include these two points together.

In ‘recruitment and study processes’ the authors state participants were recruited from ‘regional, remote and urban settings where lung cancer incidence is higher than in metropolitan settings’. Can simpler language be used here to make this easier to understand, for example, use the same language as in Table 1, ‘urban/inner-city, suburban and rural’?

The rest of this paragraph explains the excellent design of the study and the writing is clear and easy to understand.

The ‘Focus group content’ paragraph states that a presentation was used which was based on the findings from some international LCS randomised controlled trials, an Australian LCS enquiry and examples of international LCS programmes – could some if this data be included in the introduction? I find myself wondering what these findings are.

Were any of the topic guides reflected upon and then altered between groups to improve?

‘Data collection’ includes a sentence, which states ‘for those who could not attend a focus group, three individual interviews were conducted.’ This is slightly misleading, do they mean three interviews per participant or three participants each having one interview? There is only one sentence about the researcher conducting the groups/interviews. I’d like to know more about their background/position and their potential effect on participants and the data. By extension, are the researchers who undertook the analysis the same as those undertaking the data collection? I’d like to see more detail on the specific researchers and their roles within the data collection and analysis. Edit: I found this at the end of the paper, but it might be nice to explain briefly here?

Results

Table 1 – why are there so many participants in ‘other’ categories, such as 14 professional roles and 16 workplace settings?

Although Table 3 is a nice summary, the quotes feel ‘disembodied’ and I would like them to be embedded within the relevant text. It was difficult to go back and forwards between the table and the text when prompted to find illustrative quotes. Some words are in quotation marks within the body of text, but I was unsure if these are quotes from the data because they didn’t include participant IDs.

I am not sure how Table 2 fits in or what it adds. Could information from the table be explained within the body of the text alongside the relevant quotes to give more context and a better flow?

Table 3:

• CFIR construct ‘complexity’, Quote 1 – please explain the Australian healthcare system somewhere and what the state and federal government divide is.

• CFIR construct ‘cosmopolitan’, - I don’t understand this quote.

• CFIR construct ‘external policy and incentives’, Quote 2 – discuss somewhere the ethical implications of offering incentives to take part in a screening programme which has potential to cause harm and the impact of incentives on fully-informed shared-decisions in healthcare.

• CFIR construct ‘readiness for implementation’, both quotes in ‘workforce and programme delivery’ and Quote 2 from ‘primary care education’ relate indirectly to the specificity of CT/LDCT (and resulting recall rate and biopsy rate). As I’ve said, I think it is important to include some review of these figures from the literature in your introduction and then you can discuss this in the context of these quotes.

Results section ‘cost’, the sentence ‘costs in the set-up of the programme …’ doesn’t make sense to me.

In ‘peer pressure’, NLST criteria is mentioned, what is this?

In ‘workforce and program delivery’, ‘telehealth’ is mentioned, what is this?

In ‘primary care engagement and education’, the opening sentence ‘Participants viewed GPs as the gatekeepers to LCS.’ I’m not sure I understand what is meant by this and the implications.

In ‘reflecting and evaluating’, AI is mentioned for the first time, it would be nice to expand upon this in the discussion in light of the rise of AI used to ease workload pressure in other areas of cancer screening. There is a brilliant quote I read elsewhere in the manuscript, which could be used here about ‘boring CT reading’.

Discussion

In paragraph four, the authors mention a pilot trial, the ‘International Lung Screening Trial’ running in Australia (among other countries), so it’s already happening?

Paragraph five suggests the need for ‘patient and professional champions’, which is a lovely phrase. The sentence about a ‘bottom up approach’ being more effective than a ‘top down approach’ could be clearer.

In strengths and limitations, the authors mention that participants more knowledgeable about LCS may have been recruited than not, why is this and what is the effect, if any, on the data?

6. PLOS authors have the option to publish the peer review history of their article (what does this mean?). If published, this will include your full peer review and any attached files.

Reviewer #1: No

Reviewer #2: No

---

## [Decision Letter · Decision Letter 1]

13 Mar 2023

PONE-D-22-30637R1“What do I think about implementing lung cancer screening? It all depends on how.” Acceptability and feasibility of lung cancer screening in Australia: the view of key stakeholders about health system factorsPLOS ONE

Dear Dr. Rankin,

Thank you for submitting your manuscript to PLOS ONE. After careful consideration, we feel that it has merit but does not fully meet PLOS ONE’s publication criteria as it currently stands. Therefore, we invite you to submit a revised version of the manuscript that addresses the points raised during the review process. As you will see, the points raised are largely typographical and therefore once you have addressed them using the usual standard (tracked and with point-by-point responses) then I should be able to approve publication of the manuscript without returning to peer review. 

We look forward to receiving your revised manuscript.

Kind regards,

Jason Scott

Academic Editor

PLOS ONE

Journal Requirements:

Reviewers' comments:

Reviewer's Responses to Questions

**Comments to the Author**

1. If the authors have adequately addressed your comments raised in a previous round of review and you feel that this manuscript is now acceptable for publication, you may indicate that here to bypass the “Comments to the Author” section, enter your conflict of interest statement in the “Confidential to Editor” section, and submit your "Accept" recommendation.

Reviewer #1: All comments have been addressed

Reviewer #2: All comments have been addressed

2. Is the manuscript technically sound, and do the data support the conclusions?

Reviewer #1: Yes

Reviewer #2: Yes

3. Has the statistical analysis been performed appropriately and rigorously? 

Reviewer #1: N/A

Reviewer #2: N/A

4. Have the authors made all data underlying the findings in their manuscript fully available?

Reviewer #1: No

Reviewer #2: Yes

5. Is the manuscript presented in an intelligible fashion and written in standard English?

Reviewer #1: Yes

Reviewer #2: Yes

6. Review Comments to the Author

Reviewer #1: Overall: I want to thank the authors for the significant changes that have been completed in this revision. They have greatly improved the manuscript. There are a few minor additional comments below, other than those I have no further comments.

Pg 3 – “and over 70% of all screen detected lung cancer would be diagnosed at an early stage” – Sorry I am not sure this makes sense upon review.

Pg 4 - “barriers to lung cancer screening at participant” – change to LCS in line with the aforementioned abbreviation

Pg 6 – “Data collection: All focus groups and interviews were conducted between February and July 2021..” – double full stop

Pg 9 – “Other 7 (8.5))” – double brackets

Pg 23 – “(FG2, Thoracic Surgeon” – missing bracket

Pg 24 – “A need to learn what hasn’t” – ‘has not’ instead of ‘hasn’t’ (sorry this was not identified initially)

Pg 26 – “(FG26, Chief Executive” – missing bracket

Reviewer #2: (No Response)

7. PLOS authors have the option to publish the peer review history of their article (what does this mean?). If published, this will include your full peer review and any attached files.

Reviewer #1: **Yes: **Dr Kate Sykes

Reviewer #2: **Yes: **Helen Elliott

---

## [Author Response · Author response to Decision Letter 1]

16 Mar 2023

Review Comments to the Author

Reviewer #1: Overall: I want to thank the authors for the significant changes that have been completed in this revision. They have greatly improved the manuscript. There are a few minor additional comments below, other than those I have no further comments.

Pg 3 – “and over 70% of all screen detected lung cancers would be diagnosed at an early stage” – Sorry I am not sure this makes sense upon review.

Response: This has been amended to read, ‘of all screen-detected lung cancer, over 70% would be diagnosed at an early stage’ (Pg 3).

Pg 4 - “barriers to lung cancer screening at participant” – change to LCS in line with the aforementioned abbreviation

Response: This has been amended to ‘LCS’ (Pg 4) as suggested.

Pg 6 – “Data collection: All focus groups and interviews were conducted between February and July 2021..” – double full stop

Response: This has been corrected (Pg 6) as suggested.

Pg 9 – “Other 7 (8.5))” – double brackets

Response: This has been corrected (Pg 9) as suggested.

Pg 23 – “(FG2, Thoracic Surgeon” – missing bracket

Response: This has been corrected (Pg 23) as suggested.

Pg 24 – “A need to learn what hasn’t” – ‘has not’ instead of ‘hasn’t’ (sorry this was not identified initially)

Response: This has been corrected (Pg 24) as suggested.

Pg 26 – “(FG26, Chief Executive” – missing bracket

Response: This has been corrected (Pg 26) as suggested.

---

## [Editor Report · Decision Letter 2]

21 Mar 2023

“What do I think about implementing lung cancer screening? It all depends on how.” Acceptability and feasibility of lung cancer screening in Australia: the view of key stakeholders about health system factors

PONE-D-22-30637R2

Dear Dr. Rankin,

We’re pleased to inform you that your manuscript has been judged scientifically suitable for publication and will be formally accepted for publication once it meets all outstanding technical requirements.

Kind regards,

Jason Scott

Academic Editor

PLOS ONE
---

## [Editor Report · Acceptance letter]

28 Mar 2023

PONE-D-22-30637R2 

“What do I think about implementing lung cancer screening? It all depends on how.” Acceptability and feasibility of lung cancer screening in Australia: the view of key stakeholders about health system factors 

Dear Dr. Rankin:

I'm pleased to inform you that your manuscript has been deemed suitable for publication in PLOS ONE. Congratulations! Your manuscript is now with our production department. 

Kind regards, 

on behalf of

Dr. Jason Scott 

Academic Editor

PLOS ONE